# Mental flexibility assessment: A research protocol for patients with Parkinson's Disease and Anorexia Nervosa

**Francesca Borghesi**[1]*, **Valentina Mancuso**[2], **Francesca Bruni**[2],
**Riccardo Cremascoli**[3], **Laura Bianchi**[3], **Leonardo Mendolicchio**[4], **Stefania Cattaldo**[5],
**Alice Chirico**[6], **Alessandro Mauro**[3,7], **Elisa Pedroli**[2,8], **Pietro Cipresso**[1,3]

1 Department of Psychology, University of Turin, Turin, Italy, 2 Faculty of Psychology, eCampus University, Novedrate, Italy, 3 Istituto Auxologico Italiano, IRCCS, Unit of Neurology and Neurorehabilitation, San Giuseppe Hospital Piancavallo, Verbania, Italy, 4 Istituto Auxologico Italiano, IRCCS, Unity of Eating Disorders, San Giuseppe Hospital Piancavallo, Verbania, Italy, 5 Istituto Auxologico Italiano, IRCCS, Laboratory of Clinical Neurobiology, San Giuseppe Hospital Piancavallo, Verbania, Italy, 6 Department of Psychology, Research Center in Communication Psychology, Universitá Cattolica del Sacro Cuore, Milan, Italy, 7 Department of Neuroscience Rita Levi Montalcini, University of Turin, Turin, Italy, 8 Department of Geriatrics and Cardiovascular Medicine, IRCCS Istituto Auxologico Italiano, Milan, Italy

* francesca.borghesi@unito.it

**Data Availability Statement:** No datasets were generated or analysed during the current study. All relevant data from this study will be made available upon study completion.

## Abstract

Mental Flexibility oscillates between adaptive variability in behavior and the capacity to restore homeostasis, linked to mental health. It has recently been one of the most investigated abilities in mental and neurological diseases such as Anorexia nervosa and Parkinson's disease, studied for rigidity or cognitive inflexibility. Patients with anorexia nervosa have rigid cognitive processes about food and weight, which leads to restrictive eating and excessive exercise. People who struggle to adapt their cognitive processes and actions to change their diet and exercise habits may have a harder time recovering from the disorder. On the other hand, research suggests that Parkinson's disease patients may have cognitive flexibility impairments that impair their ability to perform daily tasks and adapt to new environments. Although of clinical interest, mental flexibility lacks theoretical liberalization and unified assessment. This study introduces "IntellEGO" a protocol for a new, multidimensional psychometric assessment of flexibility. This assessment evaluates a person's authentic ability to handle daily challenges using cognitive, emotional, and behavioral factors. Since traditional assessments often focus on one domain, we aim to examine flexibility from multiple angles, acknowledging the importance of viewing people as whole beings with mental and physical aspects. The study protocol includes two assessment phases separated by a rehabilitation period. T0, the acute phase upon admission, and T1, the post-rehabilitation phase lasting 15 days for Parkinson's patients and 4 weeks for eating disorder patients, will be assessed. Neuropsychological performance, self-report questionnaires, psychophysiological measures, and neuroendocrine measures will be collected from Anorexia Nervosa and Parkinson's Disease patients during each study phase. The objective of this procedure is to provide clinicians with a comprehensive framework for conducting meticulous assessments of mental flexibility. This

**Funding:** Research was funded by the Italian Ministry of Health- Ricerca Corrente; and Ministero dell'Istruzione dell'Università e della Ricerca MIUR project "Dipartimenti di Eccellenza 2023–2027" to Department of Neuroscience "Rita Levi Montalcini"; and PON R&I 2014-2020 (FSE REACT-EU). The founders had no roles in the study.

**Competing interests:** The authors have declared that no competing interests exist.

framework considers emotional, cognitive, and behavioral factors, and is applicable to various patient populations.

## Introduction

Mental Flexibility (MF) is a cross-cutting characteristic that is present in various cognitive-affective processes [1–3]. This concept is commonly delineated in relation to cognitive, physiological, and emotional elements [3–10]. Cognitive flexibility pertains to executive functions, specifically attention shifting and conflict monitoring [9, 11–16]. Psychological Flexibility (PF) or Emotional Flexibility (EF), instead, refers to the individual's capability to change their behavioral patterns and their willingness to engage with emotions, thoughts, and sensations, including both desired and undesired private experiences [7, 9, 10]. Psychological and emotional flexibility are often considered synonyms in scholarly literature, and for the purpose of this article, the term "emotional" will be employed [6]. All components of cognitive and emotional flexibility exhibit a common characteristic, referred to as *adaptive variability* [3, 17]. Variability refers to the subject's ability to perceive, respond to, and comprehend events from multiple perspectives [8, 10]. Similarly, adaptability, which refers to the capacity to adapt to different contexts, situations, and challenges to accomplish a specific objective, is frequently connected with resilience and resourcefulness [18, 19]. Therefore, flexible behaviour entails the ability to adjust and modify one's actions in response to changing situations and methodologies. In addition, previous studies have investigated the cognitive and emotional dimensions in conjunction with various constructs such as emotional intelligence, creativity, beliefs, and mentalizing capacities [11, 12, 14, 20–23]. In fact, the attribute of flexibility arises from the ability to adapt to social interactions, commonly referred to as emotional intelligence. Additionally, it stems from the ability to establish novel connections between ideas and beliefs, known as creativity. Furthermore, having a mindset oriented towards problem-solving, along with the capacity for beliefs and mentalizing, contributes to the development of flexibility. On the other hand, the unfavourable aspects of flexibility encompass rigidity and impulsiveness. Rigidity refers to the inclination to establish and maintain specific cognitive, emotional, or behavioral patterns, which are consistently utilized even in situations where these patterns are no longer advantageous. Rigidity is a psychological construct that has been characterized as the inclination to form and maintain consistent patterns of behavior [24, 25]. Impulsiveness is characterized by a proclivity to engage in rapid actions without due consideration of the potential outcomes, thereby constraining an individual's capacity to effectively adapt to novel circumstances or contemplate alternative strategies [26–28]. Both problems can be indicative of specific mental health conditions as stated in the Diagnostic and Statistical Manual of Mental Disorders (DSM-5-TR) [24, 25].

In this regard, both the DSM-5-TR and the U.S. National Institute of Mental Health have encouraged psychiatrists to evaluate flexibility in a dimensional manner rather than using categorical assessment. These research domain criteria (RDoC) approach recognizes that flexibility dimensions can transcend traditional diagnostic categories and encourages the integration of multiple levels of data, ranging from genomics to neural circuits to behavior and self-report (for example, using participant-filled questionnaires). This approach aims to understand the fundamental dimensions of functioning that encompasses the entire spectrum of human behavior. The "cognitive systems" domain of the RDoC matrix encompasses constructs such as "cognitive control," "emotion flexibility," and "working memory". These constructs consist of subconstructs, namely goal selection/performance monitoring and flexible updating, which

exhibit a robust association with cognitive and emotional flexibility [29]. Cognitive and emotional rigidity or inflexibility are commonly observed characteristics of many mental illnesses [3, 16, 30–35], specifically in clinical conditions that manifest in early life, such as Parkinson Disease (PD) and Anorexia Nervosa (AN). These traits have been extensively documented in the literature with numerous studies specifically examining their presence in PD [30, 31, 33, 36, 37] and AN [38–40].

On one side, recent research has established a correlation between motor stiffness in PD and executive processes (attention shifting and conflict monitoring), which are known to be affected by cognitive procedural rigidity. On the other side, AN is distinguished by its emotional rigidity and difficulties in transitioning between cognitive or affective states. Nevertheless, quantifying this emotional-cognitive rigidity remains unattainable at present. Occasionally, it is overlooked. The lack of evaluation in this context can be attributed to the inconsistent implementation of evaluation methodologies and the difficulties in precisely defining the concept of flexibility, which is currently in the process of being developed. To assess patients with PD and AN in terms of their emotional flexibility, our study describes a comprehensive assessment protocol. This method would enable a more precise restoration of people's behavioral adaptability and their ability to engage with the external environment (including people, emotions, and situations). The analysis of flexibility can be conducted across various dimensions in order to customize therapy according to the individual needs of the subject.

## Flexibility and Parkinson Disease

PD is a progressive neurodegenerative disorder that exhibits a gradual progression in function. It is characterized by the presence of misfolded α-synuclein protein aggregates in the brain, causing cell death primarily in the substantia nigra and other brain structures [41]. Traditionally, in clinical practice, PD is diagnosed based on the presence of cardinal motor symptoms, namely bradykinesia, resting tremor, and rigidity [42]. In addition to motor symptoms, PD has been documented to manifest rigidity in cognitive and personality domains as well [30, 43]. In the context of PD, individuals affected by this condition demonstrate a notable impairment in their capacity to adapt their current behaviors in order to align with the optimal response required by the surrounding environment. This cognitive ability, commonly referred to as cognitive flexibility, seems to be compromised in PD patients. For instance, Windmann and colleagues [44] hypothesized that impaired switching abilities could potentially be attributed to set-shifting impairments. This hypothesis suggests that the active disengagement from the dominant pattern may be mentally taxing for these individuals. Set-shifting, also referred to as cognitive flexibility, pertains to the ability to effectively adapt to changing environmental demands through the flexible transition between different rules, tasks, actions, and mental sets [26, 45]. Dopamine plays a crucial role in this context. The significance of dopamine in intricate cognitive processes has been widely acknowledged in the literature [28, 46]. The prefrontal cortex of the human brain exhibits a significant presence of dopamine receptors and receives substantial dopaminergic fiber inputs from other regions within the cerebral cortex. The data obtained from PD research indicates that there are early deficits in dopamine pathways caused by the initial degeneration of dopamine neurons in the substantia nigra and ventral tegmental area. As a result, the nigrostriatal dopamine circuitry is impacted by the disease, leading to subsequent damage to the mesocortical and mesolimbic pathways [47]. In this context, it is readily comprehensible to discern the related deficit of PD, which is linked to the role of dopamine and the anatomical structures in which dopamine is located. In fact, abilities traditionally dependent on the integrity of the prefrontal-striatal loops, have been shown to be particularly

sensitive to dopamine activity [43]. The cognitive domains commonly affected in individuals with PD as a result of the dopaminergic deficit include working memory, set-shifting, planning, and selective attention. Moreover, researchers suggest dopamine modulates the ability to implement nonroutine schemata and update operations (flexibility processes) [48]. Recent studies also suggest alterations in cognition, are not independent of personality, and emotional disturbance. In particular, researchers provide evidence of a typical emotional and personality pattern of inflexibility, cautiousness, industriousness, and low impulsivity [49], considering that weak set-shifting or cognitive inflexibility results in rigid or perseverative thinking and behavior.

## Flexibility and Anorexia Nervosa

AN exhibits a similar pattern of inflexibility. Extremely low body weight, issues with one's perception of one's own weight or body image, and a strong fear of gaining weight are all symptoms of AN [50]. Numerous studies have highlighted the presence of cognitive inflexibility among individuals with AN, as evidenced by both neurocognitive assessments and self-report measures [51–54]. This attribute, in conjunction with challenges related to the processing of emotions, constitutes the principal element of AN [38, 39, 55]. The research conducted by Zhang [56] and Cuesta-Zamora and colleagues [57] provides evidence indicating that specific emotions, such as rage, fear, sadness, and joy, along with longer-lasting and persistent moods, affect dietary responses during the ingestion process. Additionally, it has been noted that rigidity in problem-solving and learning new behaviors, as well as a lack of flexibility in cognition and emotion regulation, affect the maintenance of AN symptoms. These behaviors include weight control and food consumption, fixations with calorie counting, intense exercise, difficulties interacting with others, and rigidity in problem-solving. For instance, it has been demonstrated that emotional dysregulation and low self-esteem play a significant role in the onset and maintenance of emotional disorders [40, 58–61]. Furthermore, recent studies have also found how measures of cognitive flexibility, as measured by neuropsychological tests and those with self-reports, have different impacts on recognizing cognitive inflexibility. Classic neuropsychological tests, such as the Wisconsin Card Sorting Test and Trail making tests, are discriminative in investigating set-shifting problems in advanced disease stages [62]. Self-report tests of cognitive flexibility, on the other hand, would be adapted to identifying behavioral inflexibilities in the early stages of the disease, being related more to activities of daily living [51, 63]. As a result, there is an urgent need for and identification of a multidimensional evaluation aiming at concurrently exploring the emotional-cognitive and biological elements of mental flexibility.

## Objective

The project aims to develop a cognitive affective-behavioral assessment of flexibility in PD and AN patients [62]. The overall goal is to investigate and disentangle flexibility components and their interaction, to create a multidimensional assessment. For this purpose, we will focus on patients characterized by cognitive-affective rigidity problems, such as PD and AN [30, 31, 33, 38–40]. Self-report questionnaires and neuropsychological tests are commonly employed in the evaluation of Cognitive Flexibility (CF), which refers to an individual's capacity to adapt or transition between different "cognitive sets" or strategies considering environmental changes [64]. Neuropsychological measures classified as "direct" indicate the need for direct observation of the subject's behavior during various tasks. These measures primarily focus on the cognitive aspect, specifically targeting attentional and mnemonic processes. All these theories are founded on the assumption that transitioning from processing a series of letters to processing

a series of numbers necessitates cognitive flexibility and inhibition, as the act of switching to a numerical sequence involves inhibiting the previously processed letter sequence. Self-report questionnaires, on the other hand, are grounded in the assessment of communicative and emotional competencies, interpersonal dynamics, and adaptable emotion regulation strategies. These strategies involve adopting a multifaceted perspective, employing cognitive reframing techniques to manage thoughts and emotions, contextualizing the situation to regulate internal states, and employing cognitive restructuring to mitigate the adverse impact of negative thoughts and emotions. According to a recent meta-analysis, the self-reports and neuropsychological tests assessing "flexibility" did not measure the same underlying construct of flexibility [13, 65]. Furthermore, self-report measures are indirect and retrospective methods of measurement, involving the investigation of emotional behavior by asking individuals to rate a series of statements. Therefore, a dearth of direct quantification of emotional flexibility exists. Consequently, we opted to assess emotional flexibility through direct behavioral analysis utilizing the International Affective Picture System (IAPS). This involved measuring various physiological parameters such as electrocardiogram (ECG), blood volume pulse (BVP), galvanic skin response (GSR), respiration pattern (RSP), facial zygomatic electromyography (EMG-Z) and corrugator (EMG-C), and electrooculography (EOG) [17, 66, 67]. The IAPS picture set has been specifically created to elicit emotional responses. The images within the set are categorized as pleasant, neutral, or unpleasant, and we used it, aiming to explore the range of affective states in a dynamic manner [68–70]. In fact, in recent studies, there has been a considerable amount of research conducted on the temporal changes in affective states, commonly referred to as affect dynamics [71–73]. The objective is to conduct a direct analysis of the transitions from one affective state to another, in order to quantify behavioral and neurophysiological variability as a measure of emotional flexibility [17, 66, 67, 74]. Mental Flexibility has as its main characteristic *adaptive variability*, which in this case comes to be regarded as affective variability.

It would therefore distinguish two major components of Mental Flexibility, cognitive and emotional, which would be examined in both direct (e.g., neuropsychological tests and IAPS) and indirect (e.g., self-reports) terms.

Furthermore, this study aims to assess endocrinological parameters, including interleukin 6, cortisol, serotonin, catecholamines, and endorphins, in individuals diagnosed with eating disorders. These assessments will be conducted at three time points: the beginning of the rehabilitation program (T0), the midpoint of the rehabilitation program, and the conclusion of the rehabilitation program (T1). The selection of these parameters is based on their established correlations with cognitive and behavioral flexibility [3, 75]. For example, previous studies have demonstrated that a temporary reduction in cerebral serotonin levels can impact the cognitive processing of negative feedback in the context of reversal learning [76–78]. Moreover, the utilization of psychostimulants such as methylphenidate, which exert an influence on the activity of dopamine and noradrenaline, has been linked to enhanced attention but possible impediment to cognitive flexibility. In experiments conducted on rhesus monkeys administered therapeutic doses of methylphenidate, researchers have noted a potential negative impact on task-switching abilities [3]. These findings suggest that increased attentional focus may be accompanied by a reduction in cognitive flexibility. Furthermore, it has been observed that the striatal cholinergic systems play a role in influencing behavioral flexibility. Studies conducted on humans using proton magnetic resonance spectroscopy have revealed that a decrease in choline levels within the dorsal striatum is linked to a reduced number of perseverative trials during reversal learning. Previous studies have demonstrated that an excessive expression of IL-6 has the potential to cause cognitive dysfunction by negatively affecting neurotransmission in brain regions that regulate cognitive processes, including the hippocampus and prefrontal

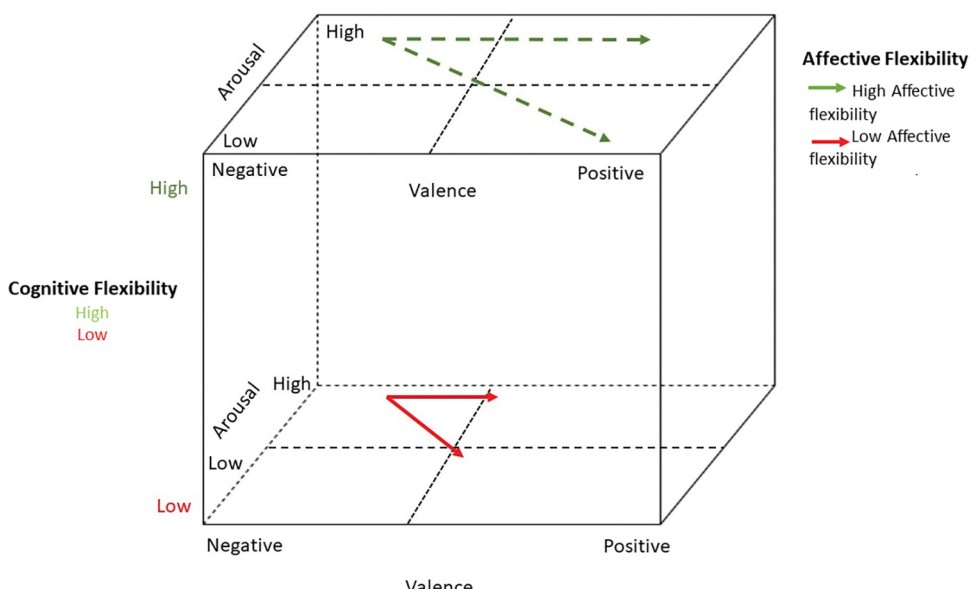

**Fig 1. Cognitive and affective components of mental flexibility interaction.**

cortex. The dysregulation of neurotransmission has the potential to impair cognitive flexibility by causing difficulties in set-shifting and attention [76–80].

Using the proposed multidimensional assessment has the dual objective of investigating the affective, cognitive, and neurobiological components of MF, and its relationships among the different components. In general, people who exhibit greater cognitive rigidity are believed to struggle to change their emotional states: individuals who exhibit higher levels of cognitive rigidity, as assessed through self-report questionnaires and neuropsychological assessments, may demonstrate reduced variability in emotional responses (Fig 1).

Through the process of systematization and analysis, one can gain insight into the cognitive-affective and physiological elements of flexibility. This allows for a comprehensive understanding of the domains that may be most affected and the potential interconnections that exist among these domains. The evaluation of MF presents a novel and intricate task, necessitating the examination of the interplay between its various constituents.

Furthermore, we will collect also emotional and cognitive variables related to mental flexibility (Table 1):

i. It is predicted that lower emotional regulation scores, emotional intelligence scores, and higher levels of ataxia, anxiety, and depression, as determined by concurrent tests, will be associated with lower levels of flexibility (cognitive and emotional component).

**Table 1. Concurrent and divergent measures of mental flexibility.**

| Higher mental flexibility | Increase Emotion regulation abilities | Increase Emotional Intelligence | Increase Creativity | Decrease Impulsiveness | Decrease Anxiety and depression |
|---|---|---|---|---|---|
| **Lower Mental flexibility** | Decrease Emotion regulation abilities | Decrease Emotional Intelligence | Decrease Creativity | Increase Impulsiveness | Increase Anxiety and depression |

Furthermore, in previous research, it has been shown that there is a strong negative correlational association between low levels of mental flexibility and high levels of impulsivity. So, we expect that high levels of impulsivity will correspond to low levels of mental flexibility.

ii. Higher levels of flexibility are expected to be associated with better emotion regulation, higher level of emotional intelligence and creativity, as well as lower levels of ataxia, anxiety, and depression and impulsiveness.

## Method

### Participants

Patients with AN according to DSM-5-TR Diagnostic Criteria [50] aged between 18–45 and patients with PD according to 2015 Movement Disorder Society Clinical Diagnostic Criteria [81] with years range between 45–80 will be recruited. The protocol was approved by the Ethical Committee of the hospital IRCCS Istituto Auxologico Italiano.

For both groups of patients, the following exclusion criteria will be identified:

i) presence of psychiatric pathologies (bipolar disorder, psychotic symptoms, history of psychosis);

ii) acute infectious diseases (including infections affecting the respiratory or genitourinary system);

iii) chronic inflammatory diseases (including Chron's disease, and ulcerative colitis);

iv) other diseases of the central nervous system (epilepsies, inflammatory or vascular diseases of the central nervous system, genetic diseases affecting the central or peripheral nervous system);

v) other medical pathologies in the phase of decompensation that could constitute a confounding factor for the purpose of the study (e.g., tumors, HIV, diabetes mellitus, cardiovascular diseases);

vi) tachyarrhythmias, bradyarrhythmias, or other heart rhythm disorders that could compromise the study of heart rate variability.

The last two exclusion criteria apply only to patients with AN:

vii) pregnant women;

viii) lactating women.

### Sample size

The sample size was calculated using Gpower 3.1, separately for PD and AN population class. For eating disorders, the study by Sierra and colleagues [82] was used to estimate effect dimension: in the study, IAPS (International Affective Picture System) results were used as a variable dependent on predictive variables describing the emotionality of patients, such as the PANAS (Positive and Negative Affect Schedule). In order to be more conservative and with an effect size of 0.34 and a statistical power of 0.95, we project about 40 subjects, considering eventually drop-out, for the patient and control groups, respectively.

**F tests**—ANOVA: Repeated measures, within-between interaction

**Analysis:** A priori: Compute required sample size

**Input:**    Effect size f        = 0.34

α err prob    = 0.05
Power (1-β err prob)   = 0.95
Number of groups   = 2
Number of measurements  = 2
Corr among rep measures  = 0.5
Nonsphericity correction ε  = 1

**Output:** Noncentrality parameter λ  = 14.7968000
Critical F    = 4.1708768
Numerator df   = 1.0000000
Denominator df   = 30.0000000
Total sample size   = 32
Actual power   = 0.9608705

The study by Ille and colleagues [83] was used to estimate the effect dimension of PD group: The study used IAPS to investigate patients' emotional reactions. In order to be more conservative and with an effect size of 0.39 and a statistical power of 0.95, we project about 30 subjects, considering eventually drop-out, for the patient and control groups, respectively.

**F tests**—ANOVA: Repeated measures, within-between interaction

**Analysis:** A priori: Compute required sample size

**Input:** Effect size f   = 0.39
α err prob    = 0.05
Power (1-β err prob)   = 0.95
Number of groups   = 2
Number of measurements  = 2
Corr among rep measures  = 0.5
Nonsphericity correction ε  = 1

**Output:** Noncentrality parameter λ  = 14.6016000
Critical F    = 4.3009495
Numerator df   = 1.0000000
Denominator df   = 22.0000000
Total sample size   = 24
Actual power   = 0.9544574

Control groups will be paired by age and gender relative to patient groups.

Top of Form

## Materials and procedure

The study protocol includes two phases of assessment, spaced out with a rehabilitation period. Specifically, the assessment will be conducted at T0 (acute upon admission) and T1 (post-rehabilitation lasting 15 days for Parkinson's patients and 4 weeks for patients with eating disorders). In each phase, we will collect data on neuropsychological performance, self-report questionnaires, psychophysiological measures, and neuroendocrine measures for participants with AN, as described in Fig 2. In some cases, the tests for the two populations will be different, for example, in PD patients the administration of the tests could be leaner and faster (Fig 2).

The neuropsychological tests and self-report questionnaires will be administered to patients at their admission to acute care (administered by T0-T1, all with an Italian validation and adaptation). Following, the IAPS will be administered at both times T0 and time T1. Images with stimuli that are too salient in comparison to the disorders under study will be eliminated using the arousal-valence selection criteria (i.e. food for patients with an eating disorder). The order of administration will be planned and distributed randomly among the subjects. This set

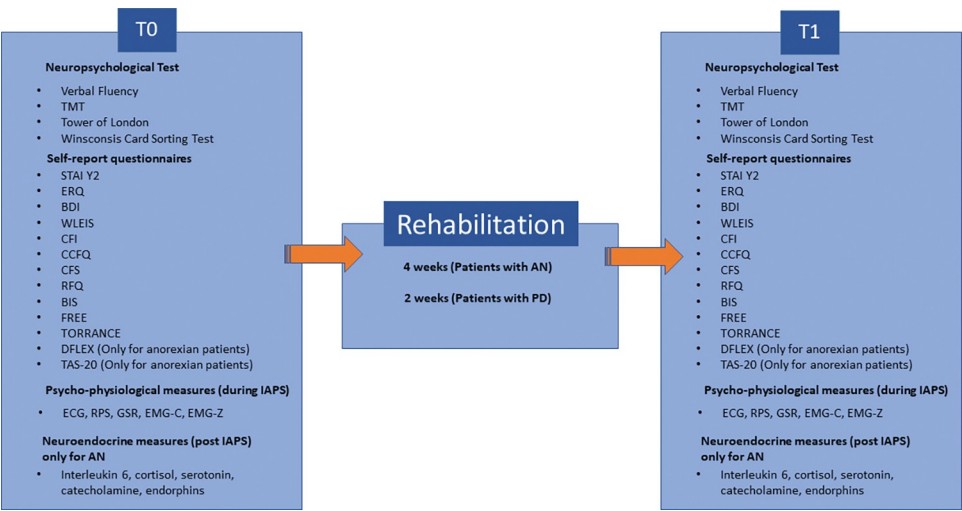

**Fig 2. Structure of assessment.**

comprised 156 images, each lasting for a duration of 10 seconds. These images were grouped into 13 blocks, to cover 12 possible transitions between quadrants (Fig 3). Each block had 12 images, each containing combinations of varying arousal and valence levels. The IAPS images had standardized scores of Arousal and Valence, according to the 9-likert point of Self-Manikin assessment (SAM). We choose only images with those specificities: for high arousal/valence (>6 Likert point scores), and for low arousal/valence (<4 Likert point scores). Notably, each participant's sequence of image presentation was randomized to minimize order effects.

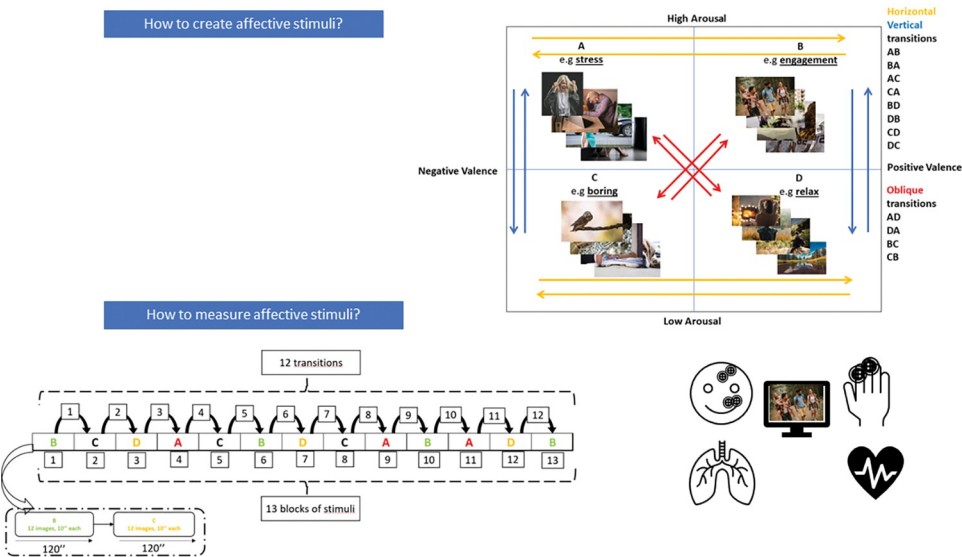

**Fig 3. IAPS protocol.** The IAPS protocol first involves careful selection of images consistent with the affective states quadrant (top part of the figure) and then implementation into a sequence consisting of 13 image blocks, resulting in 12 possible transitions for analysis. Emotional elicitation is then measured in physiological terms by f-EMG, GSR, ECG, RSP. The stimuli images shown are for illustration purposes only and are not the used ones that have been extracted from IAPS database.

Peripheral physiological parameters such as the ECG, BVP, GSR, EOG, RSP, and facial EMG (corrugator and zygomatic) will be detected throughout the test as measurable indicators of emotional arousal (T0-T1). Additionally, morning venous blood sampling is planned to analyse biohumoral parameters in subjects with eating disorders, including interleukin 6, cortisol, serotonin, catecholamines, and endorphins at T0 and T1.

PD patients and participants affected by AN will be recruited at the IRCCS Auxologico Piancavallo, at the Neurology and Neurorehabilitation Unit, and the Rehabilitation of eating disorder and Nutrition Unit respectively. Patients will receive all the information required to sign the informed consent form, and inclusion/exclusion criteria will be evaluated by a neurologist and an expert physician in eating disorders.

### Assessment

Patients received a complete behavioral and neuropsychological evaluation to assess flexibility, divide into three parts: (i) neuropsychological tests, (ii) self-report questionnaire, and (iii) IAPS administration. The assessment will be done in acute care (T0) and the same tests will be performed after a period of rehabilitation (T1). The battery includes the Trail Making Test **(TMT),** the Tower of London **(ToL),** the phonemic and semantic verbal fluency, the phonemic/semantic alternate fluency test, and the Modified Wisconsin Card Sorting Test **(WCST).** The TMT [84] comprised two parts: TMT-A and TMT-B. The first is a searching task, the second requires shifting abilities. The request is to connect the numbers from 1 to 25 in sequential order or numbers from 1 to 13 and letters from A to N in alternating order (i.e., 1, A, 2, B, etc.) respectively. The time used to perform each task is considered during the scoring: the highest the time the worst score. The ToL traditionally measures strategical reasoning, problem-solving, and mental planning. The task consists of moving tree beads to reproduce a target configuration; specifically, three abilities are required: (a) creating a general plan; (b) arranging beads into a series of movements; and (c) maintaining the general plan in working memory. Time and accuracy are evaluated [85]. Phonemic and verbal fluency enables a quick and effective assessment of lexical access and the breadth of semantic inventory, the ability to use research strategies in the lexicon, and spontaneous flexibility. In the phonemic part, patients are asked to report as many words as they can produce beginning with a certain letter of the alphabet (F, L, P); while, in the semantic part they are asked to report words belonging to a specific category (car brands, fruits, animals). Patients have a minute of time and the sum of the right items produced for each letter/category determines the final score. Moreover, the alternate fluency [86] is used to analyze set-shifting abilities. The task requires rapidly changing the mindset to generate words by continuously alternating between phonemic and semantic criteria. The WCST consists of sorting the cards; However, the sorting rule must be deduced based on the feedback (correct sorting or not). As soon as the rule is found, it is altered, and the task then finds a new sort rule. Perceptual criteria of the elements on the cards, such as color, shape, or number, are the basis for the sorting rules. The Modified Card Sorting Test consists of a set of stimulus and response cards. The first is composed of four cards each of which is unique in terms of color (red, green, blue, or yellow), shape (triangle, star, cross, or circle), and the number of items (one, two, three or four). The response cards consist of 2 sets of 24 cards comprising all possible combinations of the color, shape, and number attributes. The subject is instructed to place each response card below a stimulus card, deducing sorting rules based on the clinician's feedback (right or wrong placement) [87].

Similarly to the neuropsychological assessment, both groups will also perform self-reports of cognitive flexibility with the following instruments: (i) Cognitive Flexibility Inventory **(CFI)** [88, 89], with 19 items, falls into two distinct subclasses: Alternatives (12 elements) and

Control (7 elements). These items are rated on a 7-point Likert scale (1 being strongly disagreed with, and 7 being strongly agreed with. The highest results generally indicate high cognitive flexibility in the overall measure; (ii) Cognitive Control and Flexibility Questionnaire **(CCFQ)** [90] is a 18-item questionnaire that assesses a person's perceived capacity to exert control over unwanted (negative) intrusive thoughts and emotions as well as their capacity to adapt to stressful circumstances; (iii) Cognitive Flexibility Scale **(CFS)** [91]: the questionnaire consists of 12 Items that evaluate the relational communication flexibility; (iv) Flexible Regulation of Emotional Expression **(FREE)** which assesses emotional regulation in terms of flexible management between negative and positive affective states and vice versa (16 items) [92].

Patients with AN will be also assessed with additional tests: (v) Detail and Flexible Questionnaire **(DFlex)** [93] is a self-report tool for assessing cognitive inflexibility, based on cognitive styles.

Finally, we will measure also concurrent constructs, significantly associated with flexibility:

(vi) State-Trait Anxiety Inventory-subscale Trait **(STAI Y2):** It is a questionnaire made up of 20 items on a Likert scale with a four-point scale that refers to the Trait Anxiety; (vii) Emotion Regulation Questionnaire **(ERQ- 10 item)** [94, 95] is a 10-item questionnaire with two scales representing two distinct methods of emotion regulation: cognitive reassessment (6 items) and expressive suppression (4 items). The questions focus on the subject's emotional life, particularly his ability to control (i.e., regulate and manage) his emotions. The 10 items are scored using a 7-point Likert scale that ranges from strongly disagreeing to strongly agreeing; (viii) Beck Depression Inventory **(BDI)** is a self-evaluation tool with 21 multiple-choice questions about the subject's depressive states; (ix) The Wong and Law Emotional Intelligence Scale **(WLEIS)** [96] is a measure of Emotional Intelligence of 16 items, based on the skill model of Mayer and Salovey. The scale measures four dimensions: Auto-Emotional Evaluation, Evaluation of Others' Emotions, Use of Emotions, Regulation of Emotions; (x) Barratt Impulsiveness Scale **(BIS)** [97] is a widely used measure of impulsivity. It includes 30 elements that are evaluated to produce six first order factors (attention, motor, self-control, cognitive complexity, perseverance and impulsiveness of cognitive instability) and three second order factors (attention impulsiveness, motor and unplanned); (xi) Reflective Function Questionnaire **(RFQ)** [98, 99] consists of 8 items that classify hyper or hypo mentalization, in terms of the ability to interpret both self and others internal mental states such as feelings, desires, goals, desires and attitudes; only for AN (xii) Toronto Alexithymia scale **(TAS-20)** is a 20-question tool that is one of the most commonly used measures of alexithymia. People with alexithymia have trouble naming and describing their emotions, tend to downplay their emotional experiences and concentrate on the outside world.

## Rehabilitation

Between the assessment phases, participants will be allocated to a rehabilitation program. It will be different based on the group: 15 days for PD and 4 weeks for anorexia patients.

PD will participate in a daily rehabilitation program from Monday to Saturday including one hour of physical therapy each day. Patients will be guided by a physiotherapist as they complete exercises that improve gait speed, walking ability, and balance. Depending on the patient's comorbidities, the additional two hours per day may involve one or more of the following activities: rehabilitation psychology, speech and swallow therapy, language therapy, cognition therapy, and therapies oriented toward daily function and community reintegration.

Instead, patients with AN will go through 4 weeks of rehabilitation based on refeeding in a medical setting. The refeeding program is implemented by a multidisciplinary team of doctors, dietitians, educators, and psychologists. The following criteria form the basis of the refeeding

strategy: Higher calorie feeding is preferred in moderately malnourished patients with AN; meal-only approaches or combined nasogastric plus meal-feeding approaches are taken into consideration based on patient compliance; higher calorie refeeding strategies appear safe in severely malnourished patients with AN in the presence of medical monitoring and electrolyte correction.

As a crucial step in assisting the refeeding process, the rehabilitation psychology program will be coordinated and supervised by the psychologist with the aid of the educator, with daily individual meetings.

Patients who can support physical activity will go through a standardized rehabilitation program: during these 4 weeks, they will be directed by a physiotherapist through exercises to improve their walking ability and endurance, and balance control an hour a day, from Monday through Saturday. The proposed rehabilitation is an element of clinical routine, and we need to understand whether and how mental flexibility, encoded in its cognitive-affective dimensions, is a trait or state dimension. Patients, after routine rehabilitation performed in the hospital, might have an improvement in the affective-cognitive components of MF [53].

## Timing of the study

The study will take two years to complete, with patient recruitment and data collection taking 18 months, and data analysis and processing taking 6 months. These timelines account for the time required to: enroll patients; administer assessment scales; provide effective therapeutic care; data collection and data analyzing.

## Analysis of biomakers

All endocrinological measures are analyzed in IRCCS Auxologico Piancavallo. Patients will be fasted in the morning between 7.30 and 8.30 am with a sample of 2 serum tubes and 3 plasma tubes for the dosage of interleukin 6, cortisol, serotonin, catecholamines and endorphins.

The samples will be centrifuged and aliquoted at -80 degrees Celsius in order to preserve the analytes that will be dosed. A kit from the company IBL-International called an ELISA will be used to measure the dosages of serotonin, catecholamines, and endorphins. Endocrinological detection will take place on days other than rehabilitation protocol.

## Data collection

The patient data sheet will be filled out with clinical and disease information, including results from instrumental and humoral tests (electronic CRF). Based on the center code and enrollment sequence, an alphanumeric code will be used to identify patients (e.g., PI-001).

## Statistical analysis

The statistical methodology employed in this study centers around a 2x2 univariate analysis of variance (ANOVA) model. In this model, the between-subjects factor encompasses the distinction between patients and controls, while the within-subjects factor is delineated by the acute phase upon hospitalization (T0) and post-rehabilitation (T1). The primary objective of adopting a univariate approach is to systematically assess the effects of both group membership and time on measures of cognitive flexibility and affective flexibility. Furthermore, we seek to investigate potential interaction effects that may exist between these factors (Fig 4).

For assessing Cognitive Flexibility, which involves a diverse array of measurements, we plan to construct a composite score solely relying on self-report data. This composite score will be derived through an Exploratory Data Analysis (EDA) approach employing linear

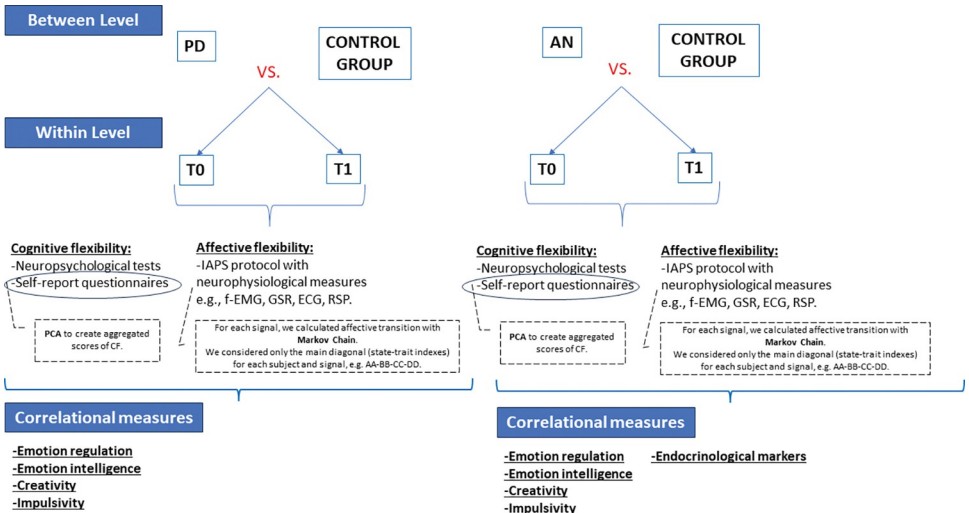

**Fig 4. Analysis framework.** For each measure of cognitive and affective flexibility an ANOVA 2*2 is implemented with between level of the group (PD/AN vs. Control Group) and within level of time (T0 vs. T1). Then a correlational analysis will be run between cognitive and affective components of flexibility and concurrent measures, considering both T0 and T1 measures.

principal component analysis (PCA). By applying PCA, we will extract common underlying dimensions from the various cognitive flexibility self-reports (e.g CFI, CFS, CCFQ, FREE, Dflex) allowing us to aggregate similar items from distinct scales into a weighted average score (Fig 4).

In the context of affective flexibility, we will consider transition indicators derived from physiological measuring (ECG, f-EMG, GSR, RSP) and analyzed by the Markov chain models. The choice of Markov chains as our analytical framework stems from their inherent capacity to capture probabilistic transitions between discrete states, a quality that aligns seamlessly with the nuanced shifts and fluctuations observed in affective experiences. By harnessing the capabilities of Markov chains, we dissect the temporal evolution of emotional states and elucidate the underlying patterns that define affective transitions. Our experimental design affords us the opportunity to calculate and evaluate affect variability across different affective transitions, which can subsequently be normalized and integrated into the Markov chains. Initially, we will calculate the reciprocal ratio between the absolute mean and variability of the signal standardized variability measures based on the inverse of Noise to Signal. The standardized variability index is computed for each transition, considering the 30-second interval between blocks (120"+-15") (Fig 5). However, it is also necessary to consider state indexes for each block in the Markov matrix, specifically the transition of state e.g. A to itself, called *state trait transition*. State trait transitions represent the probability of remaining in the same emotional state, and in the Markov Matrix, they are represented as the main diagonal. These state trait indexes are determined during the central 30 seconds of each affective block (from 45 to 75 seconds) Within the Markov matrixes, we will relativize the prior transition variability indexes, to experiment the probability of passing from one affective state to another. Normalization of transition indexes is required because the sum of transitions in Markov matrices is 1.

By doing so, we effectively capture the probability of transitioning between and into blocks, empowering to uncover the intricacies of affect dynamics in both music's auditory realm and images' visual domain.

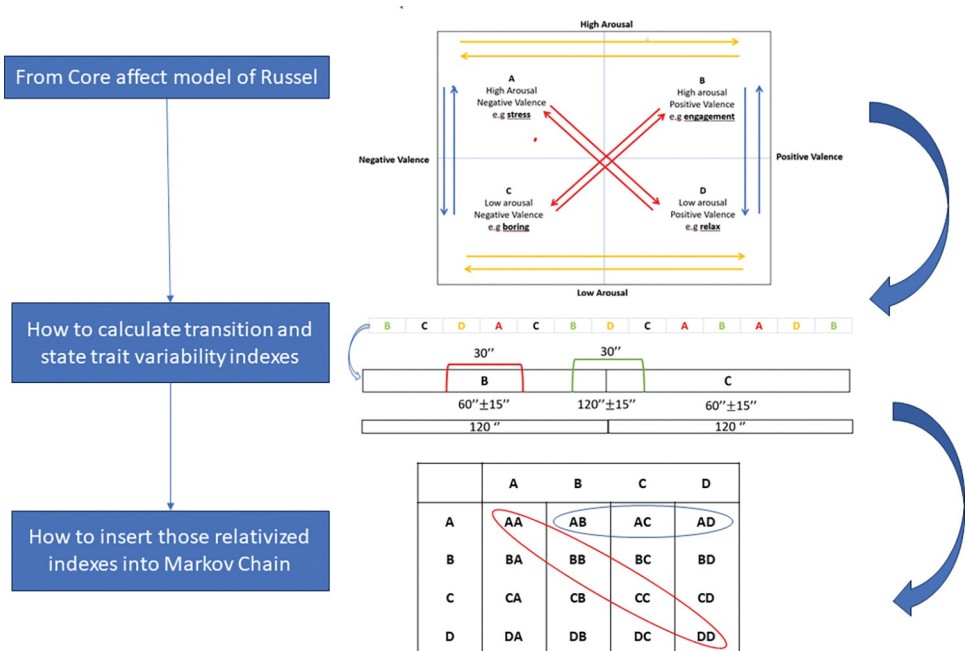

**Fig 5. Markov chain model.** From the Core affect model of Russel we can compute 12 complete transitions between affective states, we can insert them in a randomized sequence, calculate transition between blocks and state trait indexes, and once the variability indexes have been relativized, we can fit them into a Markov matrix.

From each subject's data and for each physiological signal, we extracted and analyzed only the main-diagonal indexes, denoted as state trait indexes (AA-BB-CC-DD), which convey the subject's probability of remaining anchored in a specific affective state (stability).

This meticulous methodology allows us to rigorously assess and understand the dynamics of affective experiences through the lens of Markov chains, offering valuable insights into the temporal evolution of emotional states in our experimental context.

Our anticipated outcomes encompass the following hypotheses:

1. For participants within the control group, we hypothesize that there will be no statistically significant differences between the scores obtained at T0 and T1, both for measures of cognitive flexibility and affective flexibility. This expectation suggests that individuals in the control group are likely to maintain stable levels of cognitive and affective flexibility over time.

2. Conversely, we expect an interaction effect between group membership (patients) and time (T0 to T1). This interaction effect signifies our hypothesis that cognitive flexibility measures will exhibit improvement over time among patients, leading to concurrent enhancements in affective flexibility. In particular, motor rehabilitation could have a significant impact on executive function, related to mental flexibility, while cognitive-behavioral therapy of individuals with AN could have an impact on both executive function and emotional flexibility.

In essence, our statistical analysis anticipates that patients will demonstrate increased cognitive and affective flexibility throughout the study period.

These hypotheses provide a clear framework for our statistical analysis and offer valuable insights into the expected outcomes of this multifaceted assessment of flexibility.

## Discussion

Flexibility is a crucial prerequisite in a rapidly evolving world. The world's dynamism appears to be the ideal justification for developing flexibility, and when it is lacking, pathologies appear. The main traits of some of the pathologies with the highest incidence in the world—PD for neurological disorders and AN for mental disorders—are rigidity or inflexibility, here defined as synonyms. Despite having different symptomatologies or etiologies, both cognitive and emotional dysfunctionality share some characteristics. The transition and regulation of one affective state to another, as well as challenges with attention shifting and conflict monitoring (cognitive flexibility), are particularly present in both (affective flexibility). When it comes to psychological-emotional flexibility, PD appears to be more dysfunctional than ANs. There exists a substantial body of evidence indicating that both populations exhibit cognitive inflexibility. However, a comprehensive and targeted assessment specifically addressing this issue has not been developed at present. The objective of our study is to construct a comprehensive evaluation strategy for adaptability that takes into account its cognitive, emotional, and behavioral aspects. To ensure the development of highly accurate and individualized patient rehabilitations, a comprehensive exploration of flexibility is proposed. We considered both direct measures, such as IAPS and neuropsychological tests, as well as indirect measures, specifically self-reports. Despite the comprehensive nature of the selected measures, a potential future goal could involve the development of a specialized assessment that evaluates both emotional and cognitive flexibility. One potential tool that could be utilized is virtual reality (VR) and 360˚ videos [90]. VR technology has the potential to facilitate the development of cognitive and emotional flexibility assessment tests for patients with a range of disorders, including PD and eating disorders. Specifically, the utilization of 360-degree videos within VR platforms offers a promising avenue for the creation of such tests. One potential application of VR involves the creation of a completely immersive environment in which patients are exposed to diverse situations that require cognitive and emotional adaptability. These situations may include problem-solving in a dynamic and evolving environment, navigating complex social interactions, or encountering unforeseen scenarios [93–95]. Additionally, 360-degree videos can be used to build an immersive environment in which the patient can respond to certain tasks and interact with stimuli, allowing for a realistic assessment of their cognitive and emotional flexibility [100]. When it comes to patients with eating disorders, for instance, immersive videos can be used to create a virtual environment in which the patient is exposed to foods that they ordinarily avoid or to social activities that involve eating, evaluating their emotional and cognitive flexibility in managing the situation. Additionally, they can be used to train patients with PD to have more motor flexibility, for example, by creating a virtual environment where the patient must learn new movements or handle situations that call for more flexibility. In general, the use of VR and 360-degree videos enables the creation of controlled, realistic scenarios to test and train patients' cognitive and emotional flexibility, offering a novel and efficient method of treating a variety of disorders.

## Author Contributions

**Conceptualization:** Francesca Borghesi, Valentina Mancuso, Francesca Bruni, Riccardo Cremascoli, Laura Bianchi, Alice Chirico, Pietro Cipresso.

**Data curation:** Francesca Borghesi.

**Formal analysis:** Francesca Borghesi.

**Methodology:** Francesca Borghesi, Valentina Mancuso.

**Supervision:** Leonardo Mendolicchio, Stefania Cattaldo, Alice Chirico, Alessandro Mauro, Elisa Pedroli, Pietro Cipresso.

**Writing – original draft:** Francesca Borghesi, Valentina Mancuso, Francesca Bruni, Pietro Cipresso.

**Writing – review & editing:** Francesca Borghesi, Valentina Mancuso, Francesca Bruni, Riccardo Cremascoli, Laura Bianchi, Leonardo Mendolicchio, Stefania Cattaldo, Alice Chirico, Alessandro Mauro, Elisa Pedroli, Pietro Cipresso.

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
