## [Decision Letter · Decision Letter 0]

21 Aug 2023

PONE-D-23-10949Mental flexibility assessment: A research protocol for Parkinson and Anorexia NervosaPLOS ONE

Dear Dr. Borghesi,

Thank you for submitting your manuscript to PLOS ONE. After careful consideration, we feel that it has merit but does not fully meet PLOS ONE’s publication criteria as it currently stands. Therefore, we invite you to submit a revised version of the manuscript that addresses the points raised during the review process.

We look forward to receiving your revised manuscript.

Kind regards,

Vincenzo De Luca

Academic Editor

PLOS ONE

Journal Requirements:

“No”

Reviewers' comments:

Reviewer's Responses to Questions

**Comments to the Author**

1. Does the manuscript provide a valid rationale for the proposed study, with clearly identified and justified research questions?

Reviewer #1: Yes

Reviewer #2: Yes

2. Is the protocol technically sound and planned in a manner that will lead to a meaningful outcome and allow testing the stated hypotheses?

Reviewer #1: Partly

Reviewer #2: Partly

3. Is the methodology feasible and described in sufficient detail to allow the work to be replicable?

Reviewer #1: No

Reviewer #2: Yes

4. Have the authors described where all data underlying the findings will be made available when the study is complete?

Reviewer #1: No

Reviewer #2: No

5. Is the manuscript presented in an intelligible fashion and written in standard English?

Reviewer #1: Yes

Reviewer #2: Yes

6. Review Comments to the Author

You may also provide optional suggestions and comments to authors that they might find helpful in planning their study.

Reviewer #1: The authors aim to develop an assessment framework for patients with AN and PD. While the goal is of great interest and clinical importance, there are some issues with the manuscript that should be resolved prior to publication.

1. There is no description of how the data from the different measurements is planned to be aggregated in a flexibility score or multiple flexibility subscores.

2. There is no description of any statistical analysis planned. Thus, I cannot determine whether sample size calculation is valid.

3. While the selection of psychological questionnaires and tests is well described and seems reasonable, there is a lack of justification for the biochemical markers or how these may relate to flexibility.

4. It is not stated how IL-6 and cortisol are planned to be analyzed.

(5. Since the aim of the project is to provide clinicans with an assessment framework, the manuscript would benefit from a health economic perspective, as the proposed testing procedure seems quite intense.)

Reviewer #2: The manuscript describes the planned "IntellEGO" study, funded by the Italian Ministry of Health.

The manuscript in its present form describes a very interesting and timely study and is suitable for PLOS ONE from my perspactive. Additionally, it presents the rationale behind the planned research, so that besides the description of the planned project, the character of an "opinion" and "review" paper emerges.

The topic is cognitive flexibility in Parkinson's disease and anorexia nervosa, which are also the study populations of the studies planned. The manuscript, written by many very experienced authors, has an unusual, refreshing writing style (e.g. "there is a ton of evidence" (line 461)), with several unexpected turns in the flow. While the topic is exciting and the study relevant, some small changes could make the manuscript significantly more successful:

(1) I would suggest checking whether the following information can be added:

- The manuscript does not contain any information on the planned statistical analysis, which is unusual for a protocol paper.

- No information on the study site, is it a multicentre study?

- According to the ethics protocol, the project is designed as an "observational study", but what effects do they expect from the therapy? Is the aim of the study to map therapy effects?

- Clearer information on "about 40 subjects" with such a small number of cases (line 267) and "various population classes" in line 262.

- A therapy programme that is predominantly logopaedic and designed for flexibility of movement is described precisely. Please add whether in your opinion the psychological therapy contains elements of cognitive flexibility training or not. Do you expect effects of motor training on psychological flexibility? What statistical methods will be used to measure them?

- What is the role of the control group? In what way is it included in the statistical analysis? Is it really only matched by age as line 305 suggests? Are the study groups only analysed separately? Is a comparison of the disease groups planned?

- Is medication recorded or controlled in the study? Is exercise controlled for before blood sampling?

(2) The paper could benefit from more linguistic clarity.

- Especially the abstract could benefit from a restructuring, make it clearer that it is a protocol for a planned study, mention case numbers, title of the project and design already in the abstract.

- "Rehabilitation" as a keyword?

- I don't understand figure 1 with this explanation, please add information what is meant by it

- Mention the title of the study ("IntellEGO") in the manuscript and in the abstract.

- Please introduce abbreviations beforehand (for example "AN" in line

- The manuscript should be clear about the tense, avoid past tense like "collected" in line 305.

- Anorexia and anorexia nervosa should be separated in language and content (e.g. line 138), as they are separate concepts.

- Avoid "partially" in line 491. Funding of the project as part of the main manuscript?

- Little separation of content between introduction and discussion; if necessary, move the basic aspects from the discussion to the introduction.

- Line 269ff. and 290ff. better as supplemental material

7. PLOS authors have the option to publish the peer review history of their article (what does this mean?). If published, this will include your full peer review and any attached files.

Reviewer #1: No

Reviewer #2: **Yes: **Alexander Rapp, University of Tübingen, Department of Psychiatry and Psychotherapy, Tübingen, Germany

---

## [Decision Letter · Decision Letter 1]

23 Oct 2023

Mental flexibility assessment: A research protocol for patients with Parkinson's disease and Anorexia Nervosa

PONE-D-23-10949R1

Dear Dr. Borghesi,

We’re pleased to inform you that your manuscript has been judged scientifically suitable for publication and will be formally accepted for publication once it meets all outstanding technical requirements.

Kind regards,

Vincenzo De Luca

Academic Editor

PLOS ONE

Additional Editor Comments (optional):

Reviewers' comments:

Reviewer's Responses to Questions

**Comments to the Author**

1. Does the manuscript provide a valid rationale for the proposed study, with clearly identified and justified research questions?

Reviewer #1: Yes

Reviewer #2: Yes

2. Is the protocol technically sound and planned in a manner that will lead to a meaningful outcome and allow testing the stated hypotheses?

Reviewer #1: Yes

Reviewer #2: Yes

3. Is the methodology feasible and described in sufficient detail to allow the work to be replicable?

Reviewer #1: Yes

Reviewer #2: Yes

4. Have the authors described where all data underlying the findings will be made available when the study is complete?

Reviewer #1: Yes

Reviewer #2: Yes

5. Is the manuscript presented in an intelligible fashion and written in standard English?

Reviewer #1: Yes

Reviewer #2: Yes

6. Review Comments to the Author

You may also provide optional suggestions and comments to authors that they might find helpful in planning their study.

Reviewer #1: The authors have well integrated the comments into the manuscript. In my opinion the protocol is suitable for publication. I wish the authors best of luck with with their study.

Reviewer #2: The manuscript has been significantly improved. Thank you for addressing my concerns in a clear and concise manner. From my perspective, the manuscript is now publication-ready. My best wishes for the success of your study.

7. PLOS authors have the option to publish the peer review history of their article (what does this mean?). If published, this will include your full peer review and any attached files.

Reviewer #1: No

Reviewer #2: **Yes: **Alexander Michael Rapp

---

## [Editor Report · Acceptance letter]

16 Nov 2023

PONE-D-23-10949R1 

Mental flexibility assessment: A research protocol for patients with Parkinson's disease and Anorexia Nervosa 

Dear Dr. Borghesi:

I'm pleased to inform you that your manuscript has been deemed suitable for publication in PLOS ONE. Congratulations! Your manuscript is now with our production department. 

Kind regards, 

on behalf of

Dr. Vincenzo De Luca 

Academic Editor

PLOS ONE